# HTO/Cellulose Aerogel for Rapid and Highly Selective Li^+^ Recovery from Seawater

**DOI:** 10.3390/molecules26134054

**Published:** 2021-07-02

**Authors:** Hongbo Qian, Shaodong Huang, Zhichen Ba, Wenxuan Wang, Feihan Yu, Daxin Liang, Yanjun Xie, Yonggui Wang, Yan Wang

**Affiliations:** 1Key Laboratory of Bio-Based Material Science and Technology, Ministry of Education, Northeast Forestry University, Harbin 150040, China; 1961026882@nefu.edu.cn (H.Q.); 17647643971@163.com (S.H.); bazc_@nefu.edu.cn (Z.B.); wxwang@nefu.edu.cn (W.W.); yufeihan@nefu.edu.cn (F.Y.); yxie@nefu.edu.cn (Y.X.); wangyg@nefu.edu.cn (Y.W.); 2Harbin Center for Disease Control and Prevention, Harbin 150056, China

**Keywords:** Li-ion sieve, cellulose aerogel, H_2_TiO_3_, adsorption

## Abstract

To achieve rapid and highly efficient recovery of Li^+^ from seawater, a series of H_2_TiO_3_/cellulose aerogels (HTO/CA) with a porous network were prepared by a simple and effective method. The as-prepared HTO/CA were characterized and their Li^+^ adsorption performance was evaluated. The obtained results revealed that the maximum capacity of HTO/CA to adsorb Li^+^ was 28.58 ± 0.71 mg g^−1^. The dynamic k_2_ value indicated that the Li^+^ adsorption rate of HTO/CA was nearly five times that of HTO powder. Furthermore, the aerogel retained extremely high Li^+^ selectivity compared with Mg^2+^, Ca^2+^, K^+^, and Na^+^. After regeneration for five cycles, the HTO/CA retained a Li^+^ adsorption capacity of 22.95 mg g^−1^. Moreover, the HTO/CA showed an excellent adsorption efficiency of 69.93% ± 0.04% and high selectivity to Li^+^ in actual seawater. These findings confirm its potential as an adsorbent for recovering Li^+^ from seawater.

## 1. Introduction

In recent years, with the development of new energy vehicles and portable electronic devices, the lithium battery market has expanded rapidly [1,2]. Owing to this increased demand for lithium, lithium resources have faced increasing shortages. Since seawater contains an inexhaustible supply of lithium, the recovery of lithium from seawater has received considerable attention [3,4,5,6,7]. Although the total amount of lithium contained in seawater is as much as 23 billion tons globally, its concentration in seawater is only 0.17 mg L^−1^, which is much lower than the levels of other elements in seawater, such as Na^+^, Mg^2+^, K^+^, and Ca^2+^ [8]. The conventional method of extracting ions from seawater is recrystallization, but this takes a long time, involves high energy consumption, and has a low extraction efficiency. In particular, it cannot selectively extract lithium from seawater [9,10]. However, it has been revealed that Li-ion sieves with tiny vacant sites suitable only for Li^+^/H^+^ ion exchange can specifically attract Li^+^ in alkaline aqueous solutions, which is considered as a renewable, effective, environmentally friendly, and economic approach for recovering lithium recovery from seawater [11,12,13]. Among various Li-ion sieves, H_2_TiO_3_ (HTO) is of particular interest due to its higher theoretical adsorption capacity and good chemical stability in an acidic environment [14,15,16,17,18,19]. Unfortunately, many Li-ion sieves, including HTO, exist in a state of powder particles, causing agglomeration and reducing the adsorption performance of such sieves. At the same time, the difficulty in recycling the powder has greatly impeded the application of Li-ion sieves [14,20]. Therefore, there is a need for composite technologies to overcome these drawbacks, which include granulation [19,21,22,23,24,25], membrane composition [7,26,27,28,29], fiber composition [14,30], and foaming techniques [31]. Most of the studies on these technologies focused only on improving the retained Li^+^ adsorption capacity and the long-term recyclability of composite adsorbents. However, these measures usually result in sacrificing the adsorption rate of the adsorbent [14,28,31]. Aerogel is a new material with a 3D porous network, which is usually used as a carrier for adsorption and catalysis because it can provide a convenient conductive channel for hydrated cations, which allows them to be transported rapidly inside the materials without requiring extra pumping and makes them more easily captured by the active sites [32,33,34].

Cellulose is the most widespread biomass material in nature. It is non-toxic, non-polluting, and easily degraded, making it an environmentally friendly material. CA is potentially an ideal candidate for green matrices to support various active particles for the development of novel functional composites, owing to its numerous benefits, including a large surface area, low density, high mechanical strength, 3D porous network, and excellent hydrophilicity [35]. Studies have demonstrated that cellulose hydrogel can be regenerated from cellulose solution via a simple solvent-exchange process, due to the rearrangement of hydrogen bonds [36]. Subsequently, the as-prepared cellulose hydrogel can be freeze-dried to obtain CA with excellent performance as a matrix. For example, Wan and Li et al. encapsulated γ-Fe_2_O_3_ into a 3D architecture of cellulose aerogel regenerated from an alkali/urea solution. The nanoparticles on the cellulose matrix were highly dispersed and the composite adsorbents exhibited a high adsorption rate, highly efficient Cr(VI) removal, and unique magnetism [37]. Although cellulose can be dissolved in an alkali/urea solution with low cost and recyclability, the mechanical properties of the aerogel prepared in this approach is not suitable for multiple usage [38]. Fortunately, the aerogel prepared by cellulose dissolved in the ionic liquid can endure multiple adsorption/desorption cycles, and the ionic liquid can be recycled and reused, and applied on a large scale [39,40].

Here, we propose a simple and effective strategy to prepare a new and efficient Li^+^ adsorbent (HTO/CA) through dissolution, regeneration, and freeze-drying methods, with a stable hydrophilic 3D porous network. By virtue of the notable porous network and excellent hydrophilicity of the cellulose matrix, the as-prepared HTO/CA displays a high adsorption rate, long-term recyclability, and simultaneously maintains the outstanding adsorption capacity and excellent selectivity of the HTO powder for Li^+^.

## 2. Results and Discussion

### 2.1. Characterization of HTO/CA

#### 2.1.1. XRD

Figure 1 shows the X-ray diffraction patterns of the HTO powder and HTO/CA. Notably, they exhibited similar XRD patterns, and showed clear diffraction bands, as described in the literature [41,42,43]. With the addition of cellulose, the intensities of the HTO peaks decreased. However, these characteristic peaks were still present in the XRD pattern of HTO/CA, indicating that the crystal structure of the HTO had not been destroyed during the process of preparing the HTO/CA, thereby retaining its Li^+^ adsorption ability.

#### 2.1.2. SEM-EDS

The method of direct dispersion of HTO powder in the cellulose dissolution process is simple and flexible. Figure 2 shows a 3D porous network composed of large, dense sheets with multiple layers in the interior of the CA, which is the morphology of a typical regenerated natural polysaccharide polymer aerogel [44,45]. In this case, a large amount of HTO powder could be clearly observed on the surface of the CA. Furthermore, the HTO powder had good dispersion, corresponding to the results of Ti, C, and O dispersion in the elemental mapping.

#### 2.1.3. BET

The N_2_ adsorption–desorption isotherms of the as-prepared aerogel, shown in Figure 3, reveal a distinct hysteresis loop observed in the relative pressure range of 0.8–1.0, which demonstrates an open porous network and the emergence of mesopores in the aerogel [46]. As shown in the inserted image in Figure 3, HTO powder, pure CA, and HTO/CA had pore diameters in the range of 0–30 nm, with 6, 16 and 22 nm being the most abundant pore sizes, respectively. The results indicated that, with the addition of HTO powder (m_celluose_:m_HTO_ = 4:2), the increased pore size leads to decreases in the pore volume and specific surface area of the as-prepared aerogel. The specific surface area of pure CA was 112.53 ± 7.54 m^2^ g^−1^, while that of HTO/CA was 60.38 ± 2.65 m^2^ g^−1^ and the HTO powder was 22.77 m^2^ g^−1^. The abundant pores are beneficial regarding the exposure of the surface of the HTO powder in the composite, thus providing a larger adsorption area.

#### 2.1.4. FTIR

Figure 4 shows the FTIR spectra of the pure CA, HTO powder, and HTO/CA. The FTIR spectra of both aerogels revealed bands at 3362 cm^−1^, which can be attributed to the -OH stretching vibration of the hydrogen bonds. They play a pivotal role in the process of cellulose dissolution and regeneration, and they also indicate that the CA has good hydrophilicity [47,48].

#### 2.1.5. Water Contact Angle Measurement

As reported by previous studies, the hydrophilicity of the polymer considerably impacts on the adsorption performance of the Li-ion sieves [14,20,28]. The excellent hydrophilicity of the as-prepared HTO/CA is shown in Figure 5. This result revealed that cellulose is the most hydrophilic material among the studied matrices, with a water contact angle of 10.33° ± 0.04°. The CA with excellent hydrophilicity and a 3D porous structure provides a good conductive channel for the hydrated cations, allowing them to quickly enter the aerogel and be transported efficiently [34]. This can in turn increase the accessibility of HTO to Li^+^ in the matrix.

### 2.2. Effect of HTO Loading on the Li^+^ Adsorption Capacity

Li-ion sieve content is the most important factor influencing the total amount of Li^+^ extracted by the Li-ion sieves with the matrix composites [20,23,24,30]. The retained Li^+^ adsorption capacity (% *q_e_* retained) is used to signify the percent discrepancy between the *q_e_* of the HTO powder (*q_HTO_*) and that of HTO in the CA (*q_HTO/CA_*), according to Equation (1) [30].

As shown in Figure 6, five adsorbents were prepared in this work, with different mass ratios of m_cellulose_/m_HTO_ (4:1, 4:2, 4:3, 4:4, and 0:4; i.e., pure HTO powder). The Li^+^ adsorption capacity of the aerogel increased slowly with increased loading of the HTO powder. HTO/CA at a ratio of 4:4 (m_cellulose_/m_HTO_, *w/w*) showed higher Li^+^ adsorption capacity (28.58 ± 0.71 mg g^−1^) than the other composite adsorbents. When compared with the pure HTO powder with a Li^+^ adsorption capacity of 30.44 ± 0.06 mg g^−1^, substantial retention of 93.89 ± 2.32% *q_e_* was observed for the aerogel. However, previous studies revealed that the negative effect of the matrix on the adsorption capacity of Li-ion sieves can be significant mitigated by increasing the Li-ion sieve loading [30,31,49]. This may be due to the agglomeration of small particles into larger ones with increased loading of the Li-ion sieves. These larger particles are prominent on the surface of the matrix, which reduces the blocking effect of the matrix on the active sites of the Li-ion sieves [30]. As observed in the SEM image of HTO/CA, a large amount of HTO powder adhered to the surface of the 3D macropore composed of cellulose sheets, which maximized the retained Li^+^ adsorption capacity of the HTO powder. This indicates that the inhibitory effect of CA as a matrix on the Li^+^ adsorption capacity of HTO powder was weak. HTO/CA at a ratio of 4:4 (m_cellulose_/m_HTO_, *w/w*), which demonstrated higher adsorption capacity, was selected for subsequent experiments.
(1)$$\%q_{e}\mathrm{retained}=\frac{q_{HTO/CA}}{q_{HTO}}\times 100\%$$

### 2.3. Li^+^ Adsorption/Desorption Performance of HTO/CA

Figure 7a shows the adsorption kinetics of the as-prepared pure CA, HTO/CA, and pure HTO powder. As this figure shows, Li^+^ was rapidly captured by HTO in the CA within the first hour, before achieving the *q_e_* value at 12 h. However, the adsorption equilibrium of the HTO powder was almost reached after 36 h, while pure CA did not show any Li^+^ adsorption behavior. Two kinetic models were used to explore the Li^+^ adsorption behavior in HTO powder and HTO in CA, in accordance with the following equations:(2)$$\ln\left(q_{e}-q_{t}\right)=\ln q_{e}-k_{1}t$$
(3)$$\frac{t}{q_{t}}=\frac{1}{k_{2}q_{e}^{2}}+\frac{t}{q_{e}}$$

As shown in Figure 7b,c and Appendix A, the fitting results revealed that a pseudo-second-order kinetic model (r^2^ = 0.995, 0.999) could better describe the adsorption behavior of these two adsorbents than a pseudo-first-order one. This indicates that the Li^+^ adsorption of the adsorbents is dependent on the number of accessible active sites of the HTO powder [28,42,50].

In terms of the industrial application of the adsorbent, the adsorption rate of an adsorbent is important [51]. The effect of CA was confirmed by evaluating the initial Li^+^ adsorption rate (*h*) of both adsorbents, following Equation (4).
(4)$$h=k_{2}\times q_{e}^{2}t\to 0$$

According to the results, HTO/CA had a higher value of h than the HTO powder (see Appendix A). Moreover, the derived *k_2_* values also suggested that the Li^+^ adsorption rate of HTO in CA (17.38 × 10^−4^ g mg^−1^ min^−1^) was nearly five times that of the HTO powder (3.52 × 10^−4^ g mg^−1^ min^−1^). These results can be attributed to the high hydrophilicity and 3D porous network in CA.

### 2.4. Li^+^ Adsorption Isotherm on HTO/CA

A certain amount of HTO/CA (mass ratio of 4:4) was added to various 160 mL Li^+^ solutions, with Li^+^ concentrations ranging from 0 to 100 mg L^−1^. Each sample was left to stand at 25 °C, 35 °C, and 45 °C for 24 h to reach adsorption equilibrium. Subsequently, a preferential analysis was conducted to obtain the final Li^+^ concentration in the solution. The results were then applied to the Langmuir and Freundlich models to investigate the Li^+^ adsorption isotherm, using Equations (5) and (6), respectively.
(5)$$\frac{C_{e}}{q_{e}}=\frac{C_{e}}{q_{m}}+\frac{1}{q_{m}\times K_{L}}$$
(6)$$\ln q_{e}=\ln K_{F}+\frac{1}{n}\ln C_{e}$$
where *q_e_* (mg g^−1^) and *q_m_* (mg g^−1^) are the Li^+^ adsorption equilibrium capacity and theoretical maximum monolayer Li^+^ adsorption capacity of HTO in HTO/CA, respectively; *K_L_* (L mg^−1^) represents the Langmuir adsorption equilibrium constant; and *k_F_* and 1/n are the adsorption equilibrium constant and intensity of the concentration effect on adsorption, respectively.

Figure 8a shows the results of the Li^+^ adsorption equilibria of HTO/CA. The isotherm parameters were deduced by fitting the data in Figure 8b,c, and these parameters are summarized in Appendix A. It can be observed that the adsorption capacity of the adsorbent increases with temperature, which is consistent with the findings reported in the literature. This illustrates that the adsorption process in the adsorbent is an endothermic reaction [52,53].

A comparison of the correlation coefficients (*r*^2^) indicated that the Langmuir model could better describe the adsorption behavior of the adsorbent, rather than the Freundlich model. This illustrated that similar energy was required during the Li^+^ adsorption at all sites on HTO/CA. As shown in Appendix A, the maximum Li^+^ adsorption capacity of Li-ion sieve powder after formation of the composite was 28.61 mg g^−1^.

A comparison of the Li^+^ adsorption performance by the as-prepared adsorbent relative to that of other Li-ion sieve composites is summarized in Figure 8d. HTO embedded in CA exhibited a relatively high adsorption capacity and adsorption rate. On the other hand, other composites comprising Li-ion sieves typically only exhibit one of the above-mentioned advantages of HTO/CA, and so are not suitable for industrial application.

### 2.5. Selectivity Performance

The size of the ion exchange sites and the energy required for the dehydration of hydrated ions determine the selectivity of Li-ion sieves for recovering Li^+^ from seawater [41]. To confirm that HTO/CA has the same Li^+^ selectivity as the HTO powder, cation competition experiments were carried out to estimate the effect of the CA matrix on the Li^+^ adsorption selectivity of the HTO, as observed in Figure 9. Meanwhile, the capacities of HTO/CA for adsorbing Li^+^, Mg^2+^, Ca^2+^, Na^+^, and K^+^ were 3.719 ± 0.053, 0.183 ± 0.013, 0.093 ± 0.045, 0.019 ± 0.015, and 0.024 ± 0.011 mmol g^−1^, respectively, which are similar to those of HTO powder. These results suggest that the as-prepared HTO/CA possesses an excellent cation sieving effect toward Li^+^.

### 2.6. Cyclic Adsorption/Desorption Performance

Cyclic adsorption/desorption performance was analyzed to investigate the durable performance in terms of performance consistency and the structural stability of HTO/CA and the importance of loading. As shown in Figure 10a, the amount of Li^+^ adsorbed by HTO/CA only changed slightly during the first five cycles. Furthermore, after five cycles, the final Li^+^ adsorption capacity of HTO/CA was still maintained at 22.95 mg g^−1^. There are two possible explanations for explain this: (1) the extraction of Ti^4+^ from HTO when the HTO/CA was treated with acid during Li^+^ recovery [41]; and (2) the detachment of HTO from the CA due to loosening of the structure of cellulose after five adsorption/desorption cycles under alkaline and acidic conditions. In general, owing to the excellent chemical stability of HTO, the extraction of Ti^4+^ can be negligible [43]. As shown in Figure 10b, the amount of Li^+^ absorption by the HTO powder was better than HTO/CA in the first cycle, but the weight of the powder reduced significantly as the number of cycles increased because of the difficulty in recovering the weight changes of the HTO powder after each adsorption cycle, as can be seen in Appendix A.

### 2.7. Performance in Seawater

Among previous studies involving the use of seawater, simulated seawater or actual seawater adsorption experiments were indispensable, as it is important to evaluate the potential of an adsorbent for practical applications [28,34]. Based on the aforementioned experimental results, the Li^+^ adsorption efficiency, namely, η (%), was used to assess the adsorption performance of HTO/CA in seawater, according to Equation (2) [55]. The results showed that HTO/CA exhibited a Li^+^ adsorption efficiency of more than 69.93 ± 0.04% in seawater with a Li^+^ concentration in the range of 209 to 3658 μg L^−1^. This indicates that Li^+^ can be effectively recovered from seawater without the need for pH adjustment (inserted image in Figure 11a). As shown in Figure 11b, the adsorption of other cations in actual seawater by HTO/CA (including ppm level and ppb level) was negligible compared with that of Li^+^. These results indicate that HTO/CA possesses tremendous potential for the selective recovery of Li^+^ from seawater.

## 3. Materials and Methods

### 3.1. Materials

All chemicals were used without further purification. Lithium carbonate (Li_2_CO_3_, 99%), anatase-type titanium dioxide (TiO_2_, 99%), lithium chloride (LiCl, 99%), potassium chloride (KCl, 99%), sodium chloride (NaCl, 99.5%), calcium chloride (CaCl_2_, 99%), and magnesium chloride (MgCl_2_, 99%) were obtained from Sigma-Aldrich Co. Ltd. (Shanghai, China). 1-Butyl-3-methyl imidazole chloride ([Bmim]Cl 99%) was purchased from Chengjie Chemical Co. Ltd. (Shanghai, China). Hydrochloric acid (HCl, 37%), ammonium hydroxide (NH_3_·H_2_O, 25%), and ethanol (99%) were sourced from a local supplier. The method for preparing α-cellulose based on *Populus tomentosa* Carr. (supplied by Henan Province Jiaozuo National Forestry Farm) was as described in our previous report [56]. The molecular weight of the α-cellulose was determined to be 127,656 [57]. Seawater was obtained from the Bohai Sea near Yingkou.

### 3.2. Preparation of the HTO Powder

We performed the synthesis of the HTO powder with reference to a previously reported method [41,42,50]. Li_2_CO_3_ and TiO_2_ at a molar ratio of 1:1 were ground for 0.5 h to obtain a homogeneous mixture. Then, this mixture was heated to 700 °C at a ramping rate of 6 °C min^−1^, after which it was kept at this temperature for 4 h in a muffle furnace. After natural cooling, a white powder precursor, namely, Li_2_TiO_3_ (LTO), was obtained. One gram of the as-prepared LTO powder was dispersed in 1 L of a 0.2 mol L^−1^ HCl solution, which was then stirred at room temperature for 24 h for the Li^+^/H^+^ ion exchange. Subsequently, the sample was filtered, washed three times with DI water, and heated to 60 °C for 4 h. The as-obtained white powder sample was H_2_TiO_3_ (HTO).

### 3.3. Preparation of HTO/CA

A schematic diagram of the preparation of HTO/CA is shown in Figure 12. α-Cellulose (2 wt%) was initially dissolved in [Bmim]Cl at 90 °C for 1 h. Subsequently, a series of HTO powders with different loadings (α-cellulose-to-HTO ratios of 4:0, 4:1, 4:2, 4:3, and 4:4, *w/w*) was added, followed by mixing thoroughly using a magnetic stirrer to obtain a homogeneous casting solution. Then, the as-obtained solution was poured into a mold, and the air bubbles were removed using a vacuum drying oven. The sample was subsequently placed in an ethanol coagulation bath for 24 h to regenerate the HTO/cellulose hydrogel. The hydrogel was then removed from the mold and washed with DI water to eliminate excess alcohol and [Bmim]Cl. The as-prepared hydrogel was then pre-frozen at −12 °C for 12 h and subsequently freeze-dried for 48 h under high-vacuum conditions (0.010 mbar) at −56 °C. Ultimately, HTO/CA was obtained.

### 3.4. Characterization of the HTO/CA

The morphology and structural characteristics of the HTO/CA were observed under a scanning electron microscope equipped with an energy-dispersive X-ray spectrometer (SEM-EDS; TM3030, Tokyo Prefecture, Japan). Brunauer–Emmett–Teller (BET; JW-BK112, Beijing, China) was used to analyze the changes in the BET surface area and pore size distribution between before and after the formation of the CA composite with HTO. The density of the as-prepared aerogel was determined from its weight and volume. The weight of the aerogel was determined using an analytical balance, while the dimensions of the aerogel were measured using a digital caliper. The crystal structures of the HTO powder and HTO/CA were analyzed using an X-ray diffractometer (XRD; D/max-2200VPC, Tokyo Prefecture, Japan). To confirm the high hydrophilicity of the cellulose-based aerogel, contact angles were measured using a contact-angle-measuring instrument equipped with a charge-device camera (OCA20, Dongguan, China). Fourier transform infrared spectroscopy (FTIR; Frontier, Perkin Elmer, Waltham, MA, USA) was applied to characterize the abundant hydroxyl groups in the CA.

### 3.5. Li^+^ Adsorption Performance Experiment

The influence of CA on the adsorption performance of the HTO powder was evaluated through various batch adsorption and desorption experiments. The Li^+^ adsorption performance of the HTO powder was compared to that of HTO in CA in terms of (1) the adsorption capacity (*q_e_*), (2) adsorption rate (*k*), (3) ion selectivity, and (4) adsorption efficiency (*η*) in seawater.

Different quantities of adsorbents were immersed in 160 mL of LiCl solution with a certain concentration, pH, and temperature. The samples were allowed to stand to ensure that the aerogels were intact, without causing any removal of the powder [58]. After being left to stand for several hours, the supernatant was investigated using inductively coupled plasma spectroscopy (ICP) to measure the Li^+^ concentration. The Li^+^ adsorption capacity of the adsorbent (*q_e_*) was calculated using Equation (7):(7)$$q_{e}=\frac{\left(C_{0}-C_{e}\right)V}{m}$$
where *q_e_* is Li^+^ equilibrium adsorption capacity; *C_0_* (mg L^−1^) and *C_e_* (mg L^−1^) represent the initial and equilibrium concentrations of Li^+^ in solution, respectively; and *V* (L) and *m* (g) refer to the volume of the solution and the mass of HTO powder, respectively.

Selective adsorption experiment: A certain quantity of HTO/CA was immersed in a solution containing a mixture of cations (Li^+^, Mg^2+^, Na^+^, K^+^, and Ca^2+^ with concentrations of 50 mg L^−1^), which was allowed to stand for 24 h at room temperature. The various cation adsorption capacities were calculated.

Cyclic adsorption/desorption performance experiment: A certain mass of adsorbent that had achieved saturation of adsorption was immersed in a 0.2 M HCl solution, which was allowed to stand for 24 h. The regeneration process was repeated 5 times to obtain the change in Li^+^ adsorption capacity of the HTO/CA and HTO powder.

Seawater adsorption experiment: The Li^+^ concentration in seawater (with an initial Li^+^ concentration of 209 μg L^−1^) was adjusted to a range of Li^+^ concentrations between 209 and 3658 μg L^−1^ by adding an appropriate amount of LiCl [55,59]. A certain quantity of HTO/CA was infiltrated into 160 mL of seawater solution and allowed to stand at room temperature for 24 h. The initial and equilibrium concentrations of the partial cations in the seawater were measured. The cation adsorption efficiency of the adsorbent in seawater was calculated using Equation (8):(8)$$\eta \left(\%\right)=\frac{\left(C_{0}-C_{e}\right)}{C_{0}}\times 100\%$$

## 4. Conclusions

In this work, an ideal adsorption carrier that comprises CA with HTO was demonstrated to be a novel and efficient adsorbent for recovering Li^+^ from seawater. The maximum Li^+^ equilibrium adsorption capacity of the HTO/CA was 28.61 mg g^−1^ at 35 °C, and the Li^+^ adsorption isotherm agreed well with the Langmuir model. Furthermore, HTO/CA exhibited ultrahigh adsorption rates compared with pure HTO powder, which can be ascribed to its 3D porous network and the high hydrophilicity of the CA. The experimental results obtained from the cation competition experiment and seawater performance experiment showed that the adsorption of other cations by HTO/CA was negligible. The capacity of HTO/CA to adsorb Li^+^ remained high after five regeneration cycles, demonstrating the good circulation behavior of the as-prepared HTO/CA.

## Figures and Tables

**Figure 1 molecules-26-04054-f001:**
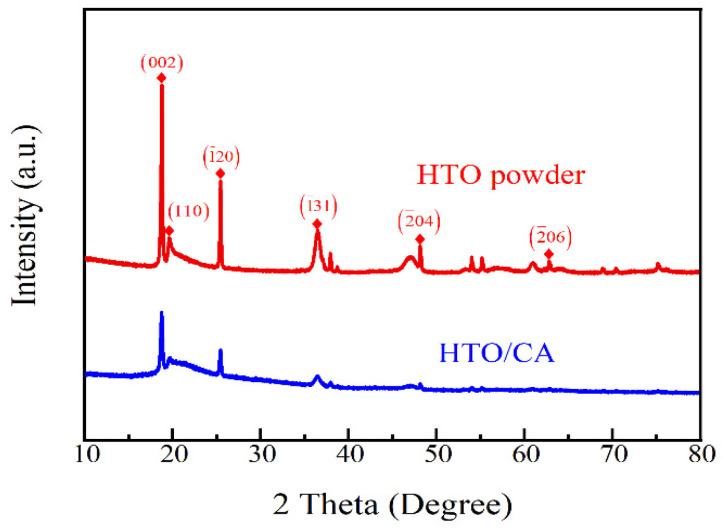
XRD patterns of the HTO powder and HTO/CA.

**Figure 2 molecules-26-04054-f002:**
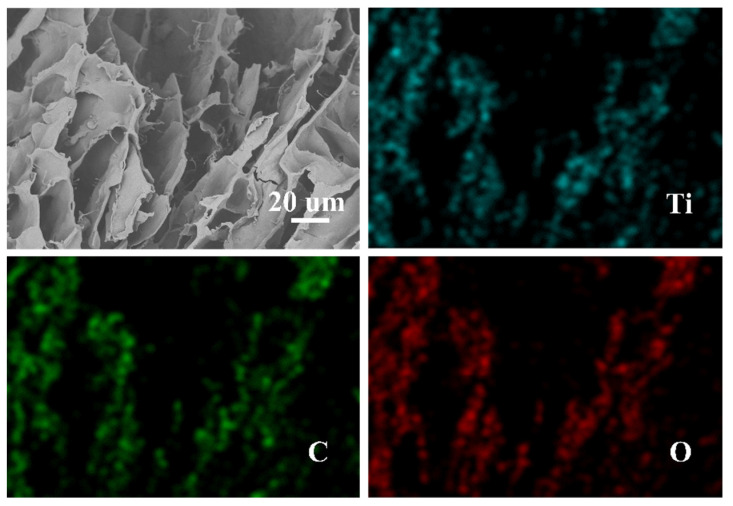
Magnified SEM images of the HTO/CA and SEM-EDS mapping of Ti, C, and O.

**Figure 3 molecules-26-04054-f003:**
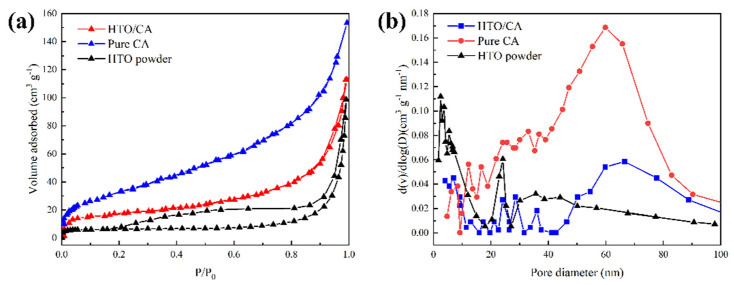
(**a**) N_2_ adsorption/desorption isotherms of the pure CA, HTO powder, and HTO/CA; (**b**) pore size distribution curves of the pure CA, HTO powder, and HTO/CA.

**Figure 4 molecules-26-04054-f004:**
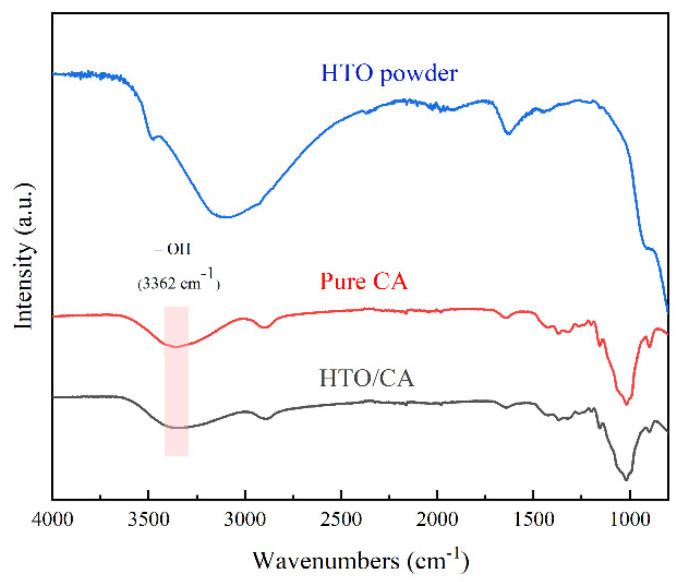
FTIR spectra of the pure CA, HTO powder, and HTO/CA.

**Figure 5 molecules-26-04054-f005:**
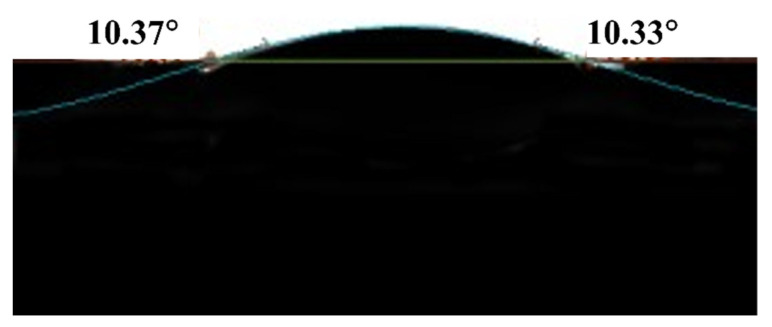
Water contact angle measurement of the HTO/CA.

**Figure 6 molecules-26-04054-f006:**
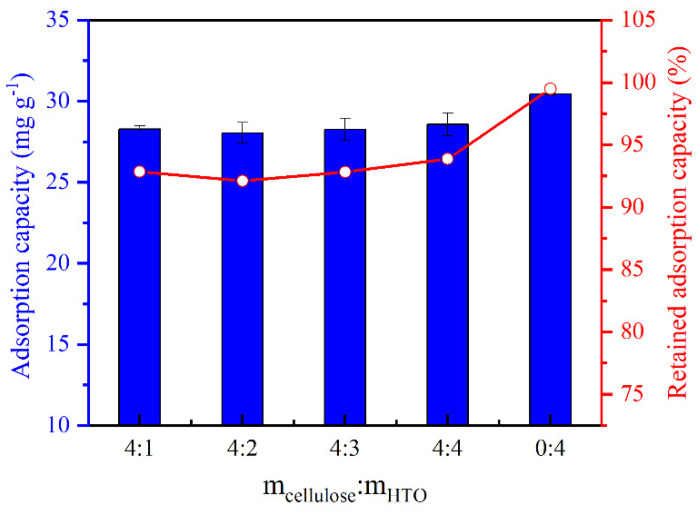
Li^+^ adsorption capacity of HTO/CA with different HTO loadings and respective % *q_e_* retained by HTO in CA for Li^+^ (pH = 10.25; V = 160 mL; m ≅ 100 mg; *C_0_* = [Li^+^] ≅ 50 mg L^−1^).

**Figure 7 molecules-26-04054-f007:**
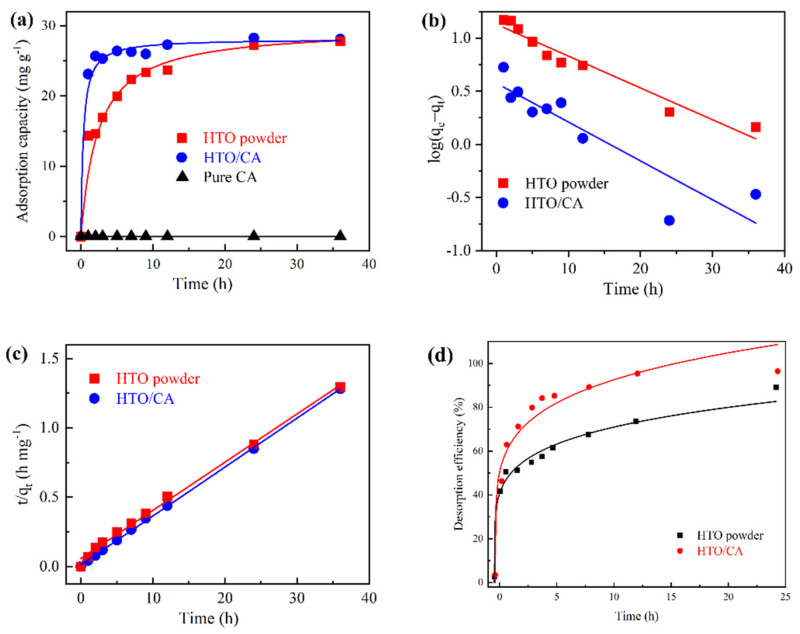
Li^+^ adsorption/desorption behavior in adsorbents: (**a**) adsorption kinetics (pH = 10.25; V = 160 mL; m ≅ 100 mg; *C_0_* = [Li^+^]≅50 mg L^−1^); (**b**) pseudo-first-order kinetics; (**c**) pseudo-second-order kinetics; and (**d**) desorption efficiency over time (V = 160 mL, *C_0_* = 0.2 M HCl).

**Figure 8 molecules-26-04054-f008:**
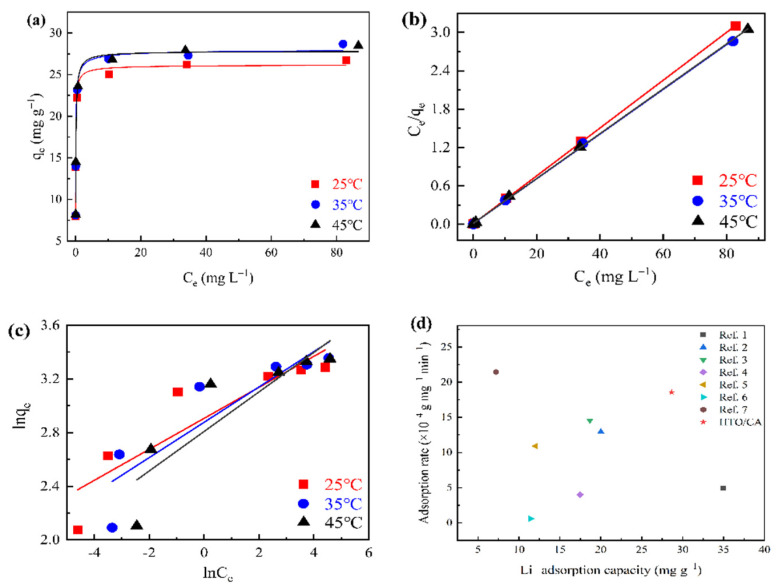
(**a**) Adsorption isotherms, and (**b**) Langmuir and (**c**) Freundlich isotherms of Li^+^ adsorption by HTO/CA at 25 °C, 35 °C, and 45 °C (pH = 10.25; V = 160 mL; m ≅ 100 mg; *C_0_* = [Li^+^] ≅ 5–100 mg L^−1^). (**d**) Comparison of the Li^+^ adsorption performance of HTO/CA with that of other Li-ion sieve composites [7,14,28,31,42,52,54].

**Figure 9 molecules-26-04054-f009:**
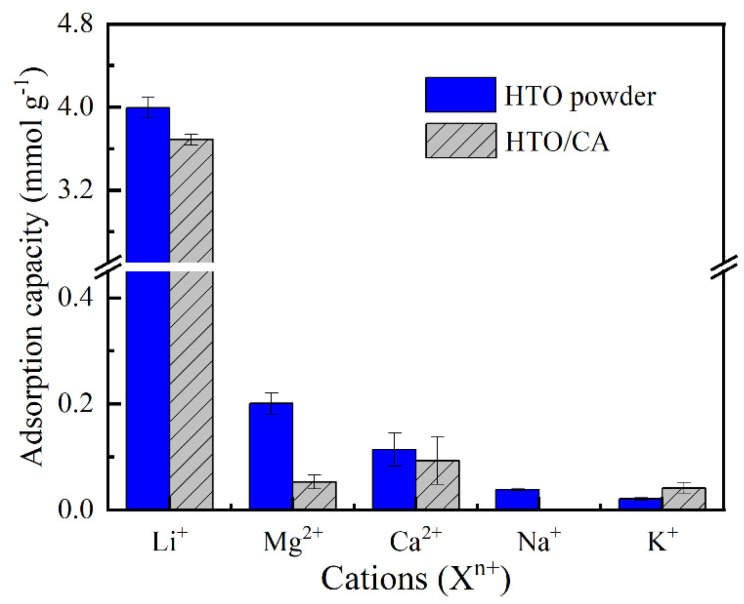
Effect of HTO loading on the selectivity of HTO/CA toward Li^+^ adsorption using a solution containing a mixture of cations (X^n+^ = Li^+^, Na^+^, Mg^2+^, K^+^, Ca^2+^) (pH = 10.25; V = 160 mL; m ≅ 100 mg; *C_0_* = [X^n+^] ≅ 50 mg L^−1^).

**Figure 10 molecules-26-04054-f010:**
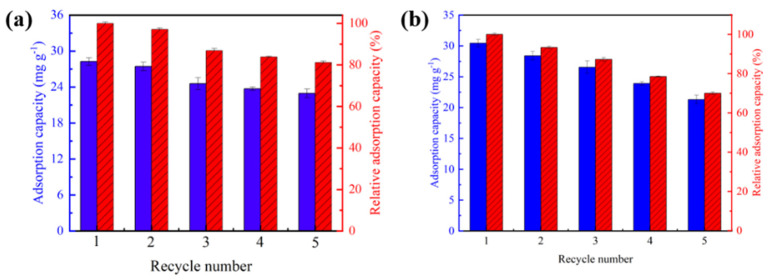
(**a**) Cyclic adsorption/desorption performance of the HTO/CA. (**b**) Cyclic adsorption/desorption performance of the HTO powder (pH = 10.25, V = 160 mL, m ≅ 100 mg, *C_0_* = [Li^+^] = 50 mg L^−1^).

**Figure 11 molecules-26-04054-f011:**
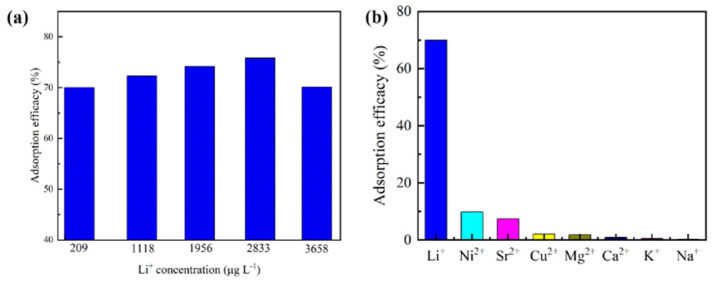
(**a**) Li^+^ adsorption efficiency of HTO/CA in seawater with different Li^+^ concentrations (*C_0_* = [Li^+^] = 209–3658 μg L^−1^, pH = 8.25, V = 160 mL, m ≅ 100 mg); and (**b**) cation adsorption efficiency of HTO/CA in seawater (concentration of cations in seawater: [Li^+^] = 209 μg L^−1^, [Ni^2+^] = 0.461 μg L^−1^, [Sr^2+^] = 0.729 μg L^−1^, [Cu^2+^] = 0.241 μg L^−1^, [Mg^2+^] = 1143 μg L^−1^, [Ca^2+^] = 457 μg L^−1^, [K^+^] = 326 μg L^−1^, [Na^+^] = 10,110 μg L^−1^; pH = 8.25, V = 160 mL, m ≅ 100 mg).

**Figure 12 molecules-26-04054-f012:**
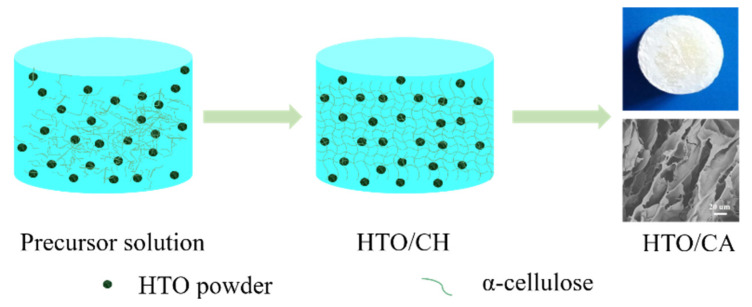
Schematic diagram of the preparation of HTO/CA.

## Data Availability

The data presented in this study are available in the manuscript and Appendix A.

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
