# Peer review of "HTO/Cellulose Aerogel for Rapid and Highly Selective Li^+^ Recovery from Seawater"

_molecules, 2021, doi:10.3390/molecules26134054_

Round 1

Reviewer 1 Report

The authors report the preparation of H2TiO3/cellulose aerogels and their use for the extraction of Li+ ions from seawater. From a general point of view, the work is well presented and of interest. The following comments should be considered by the authors :

  • figure 3 and the related text : if possible, it would be  of interest for the readers to provide the specific surface area and the pore size of pure H2TiO3.
  • figure 4 : provide FT-IR spectra from 4000 to 450 cm-1 to observed the signals of HTO.
  • figure 10 : to demonstrate that HTO/CA is of high potential for Li+ extraction from seawater, the cyclic adsorption/desorption performance of HTO/CA must be compared to that of pure HTO.

Author Response

Q1: Figure 3 and the related text: if possible, it would be of interest for the readers to provide the specific surface area and the pore size of pure H2TiO3.

(A): The specific surface area and the pore size of pure H2TiO3 have been provided and can be found in Figure 3.

Q2: Figure 4: provide FT-IR spectra from 4000 to 450 cm-1 to observe the signals of HTO.

(A): The FT-IR spectra of HTO powder was added in figure 4.

Q3: Figure 10: to demonstrate that HTO/CA is of high potential for Li+ extraction from seawater, the cyclic adsorption/desorption performance of HTO/CA must be compared to that of pure HTO.

(A): We had done the cyclic adsorption/desorption performance of HTO powder and compare with HTO/CA. Because the HTO powder is difficult to recycle, the weight changes of HTO powder after each cyclic adsorption/desorption were recorded and can be found in table S3.

Reviewer 2 Report

This is a fine paper describing a relatively simple material that is found to outperform previously described analogues in recovery of Li from sea water. The experiments are clearly described and properly executed. I would suggest to the authors to also mention that, as an ionic liquid is used for the preparation of the material, large scale application likely requires redesign and thorough optimization of the synthesis in order to reach sufficient sustainability. This is even more important given that the final applications envisaged are sustainability driven.

Author Response

This is a fine paper describing a relatively simple material that is found to outperform previously described analogues in recovery of Li from sea water. The experiments are clearly described and properly executed. I would suggest to the authors to also mention that, as an ionic liquid is used for the preparation of the material, large scale application likely requires redesign and thorough optimization of the synthesis in order to reach sufficient sustainability. This is even more important given that the final applications envisaged are sustainability driven.

(A): Thank you for your positive comment to our work. For this case, we have consulted the literature (Huang, K. L.;  Wu, R.;  Cao, Y.;  Li, H. Q.; Wang, J. S., Recycling and reuse of ionic liquid in homogeneous cellulose acetylation. Chin. J. Chem. Eng. 2013, 21 (5), 577-584. Mai, N. L.;  Ahn, K.; Koo, Y. M., Methods for recovery of ionic liquids-a review. Process Biochem. 2014, 49 (5), 872-881.  Nie, Y.;  Wang, J.;  Zhang, Z.;  Liu, X.; Zhang, X., Trends and research progresses on the recycling of ionic liquids. Chemical Industry and Engineering Progress 2019, 38 (1), 100-110.), and already started the experiment for the recovery of ionic liquid by rotary evaporation. Detailed investigation into this case may be discussed in our future publications.

Round 2

Reviewer 1 Report

All corrections were made by the reviewers. The manuscript can be accepted by Molecules.